# Structural Variability, Expression Profile, and Pharmacogenetic Properties of TMPRSS2 Gene as a Potential Target for COVID-19 Therapy

**DOI:** 10.3390/genes12010019

**Published:** 2020-12-25

**Authors:** Aleksei Zarubin, Vadim Stepanov, Anton Markov, Nikita Kolesnikov, Andrey Marusin, Irina Khitrinskaya, Maria Swarovskaya, Sergey Litvinov, Natalia Ekomasova, Murat Dzhaubermezov, Nadezhda Maksimova, Aitalina Sukhomyasova, Olga Shtygasheva, Elza Khusnutdinova, Magomed Radzhabov, Vladimir Kharkov

**Affiliations:** 1Tomsk National Medical Research Center, Research Institute for Medical Genetics, 634050 Tomsk, Russia; vadim-stepanov@medgenetics.ru (V.S.); anton.markov@medgenetics.ru (A.M.); nikita.kolesnikov@medgenetics.ru (N.K.); andrey.marusin@medgenetics.ru (A.M.); khitrinskaya@tnimc.ru (I.K.); maria.swarovskaja@medgenetics.ru (M.S.); vladimir-kharkov@medgenetics.ru (V.K.); 2Ufa Federal Research Centre of the Russian Academy of Sciences, Institute of Biochemistry and Genetics, 450000 Ufa, Russia; seregtg@gmail.com (S.L.); trofimova_nata_@mail.ru (N.E.); murat-kbr@mail.ru (M.D.); elzakh@mail.ru (E.K.); 3Medical Institute, North-Eastern Federal University, 677000 Yakutsk, Russia; nogan@yandex.ru (N.M.); aitalinas@yandex.ru (A.S.); 4Medical-Psychological-Social Institute, Katanov State University of Khakassia, 655017 Abakan, Russia; olgashtygasheva@rambler.ru; 5Laboratory of Genomic Medicine, Dagestan State Medical University, 367000 Makhachkala, Russia; radjabov_m@mail.ru

**Keywords:** TMPRSS2, ACE2, BSG, COVID-19, SARS-CoV-2, SNV, expression, pharmacotranscriptomics

## Abstract

The human serine protease serine 2 TMPRSS2 is involved in the priming of proteins of the severe acute respiratory syndrome coronavirus 2 (SARS-CoV-2) and represents a possible target for COVID-19 therapy. The TMPRSS2 gene may be co-expressed with SARS-CoV-2 cell receptor genes angiotensin-converting enzyme 2 (ACE2) and Basigin (BSG), but only TMPRSS2 demonstrates tissue-specific expression in alveolar cells according to single-cell RNA sequencing data. Our analysis of the structural variability of the TMPRSS2 gene based on genome-wide data from 76 human populations demonstrates that a functionally significant missense mutation in exon 6/7 in the TMPRSS2 gene is found in many human populations at relatively high frequencies, with region-specific distribution patterns. The frequency of the missense mutation encoded by rs12329760, which has previously been found to be associated with prostate cancer, ranged between 10% and 63% and was significantly higher in populations of Asian origin compared with European populations. In addition to single-nucleotide polymorphisms, two copy number variants were detected in the TMPRSS2 gene. A number of microRNAs have been predicted to regulate TMPRSS2 and BSG expression levels, but none of them is enriched in lung or respiratory tract cells. Several well-studied drugs can downregulate the expression of TMPRSS2 in human cells, including acetaminophen (paracetamol) and curcumin. Thus, the interactions of TMPRSS2 with SARS-CoV-2, together with its structural variability, gene–gene interactions, expression regulation profiles, and pharmacogenomic properties, characterize this gene as a potential target for COVID-19 therapy.

## 1. Introduction

The severe acute respiratory syndrome coronavirus 2 (SARS-CoV-2) has caused a pandemic of coronavirus disease (COVID-19) that has led to a global public health crisis. Infection of human cells with viral particles occurs through the binding of viral spike proteins to the receptors of the host cell and their subsequent priming with proteases. Angiotensin-converting enzyme 2 (ACE2) is considered to be the classic receptor for SARS-CoV-2, but there is evidence that the virus can also use the Basigin (BSG) receptor (CD147) [1]. The priming of viral proteins is carried out by the TMPRSS2. No specific therapy has yet been developed for SARS-CoV-2. However, blockers of all three proteins have been shown to prevent cell infection [2].

In addition to protein blocking, there are other mechanisms that can alter the expression levels of specific proteins or the affinity of their interactions with viral particles. Possible causes of expression differentiation include alterations of protein structure due to genetic variants: single nucleotide variants (SNVs), indels, copy number variations (CNVs), variants affecting regulatory regions (expression quantitative trait loci; eQTLs), and epigenetic regulation: methylation, microRNAs (miRNAs).

The TMPRSS2 gene in humans encodes a transmembrane protein of the serine protease family. Alternatively spliced transcript variants encoding different isoforms have been found for this gene. The TMPRSS2 protein is involved in prostate carcinogenesis via overexpression of Erythroblast Transformation Specific transcription factors (ETS), such as ERG and ETV1, through gene fusion. TMPRSS2–ERG gene fusion, which is present in 40–80% of prostate cancers in humans, is a molecular subtype that has been associated with predominantly poor prognosis [3,4].

The TMPRSS2 protease proteolytically cleaves and activates glycoproteins of many viruses, including spike proteins of human coronaviruses 229E (HCoV-229E) and EMC (HCoV-EMC), and the fusion glycoproteins of Sendai virus, human metapneumovirus, and human parainfluenza 1, 2, 3, 4a, and 4b viruses [2,5,6,7,8,9,10]. Both the coronavirus responsible for 2003 SARS outbreak in Asia (SARS-CoV) and SARS-CoV-2 are activated by TMPRSS2 and can thus be inhibited by TMPRSS2 inhibitors [2]. Here, we report the genetic variability of the TMPRSS2 gene in 76 human populations of North Eurasia in comparison with worldwide populations, and analyze the data with respect to the expression and regulation of TMPRSS2, its interactions with SARS-CoV-2 receptors, and its pharmacogenetic properties.

## 2. Materials and Methods

### 2.1. Structural Variability Data

Allele frequency for worldwide populations were downloaded from the GnomAD database, which contains information on the frequencies of genomic variants from more than 120,000 exomes and 15,000 whole genomes [11]. These data were used to search for SNVs and indels in the TMPRSS2 gene. CNV data were obtained from the CNV Control Database [12].

Data on allele frequencies in 76 populations of North Eurasia were extracted from our own unpublished dataset of population genomics data obtained by genotyping using Illumina Infinium genome-wide microarrays. In brief, 1836 samples from 76 human populations were genotyped for 1,748,250 SNVs and indels using a Infinium Multi-Ethnic Global-8 kit. The populations represent various geographic regions of North Eurasia (Eastern Europe, Caucasus, Central Asia, Siberia, North-East Asia) and belong to various linguistic families (Indo-European, Altaic, Uralic, North Caucasian, Chukotko-Kamchatkan, Sino-Tibetan, Yeniseian). DNA samples were collected with informed consent and deposited in the DNA bank of the Research Institute for Medical Genetics, Tomsk National Medical Research Center, Tomsk, Russia, and the DNA bank of the Institute of Biochemistry and Genetics, Ufa Federal Research Centre of the Russian Academy of Sciences. The study was approved by the Ethical Committee of the Research Institute for Medical Genetics, Tomsk National Medical Research Center. Data on four missense mutations in the TMPRSS2 gene were extracted from the dataset. The CNV search was performed using a Markov model algorithm for high-resolution CNV detection in whole-genome implemented with the PennCNV tool [13]. To determine the possible functional impact of the detected SNVs, the Polymorphism Phenotyping v2 (PolyPhen-2) tool was used [14]. PolyPhen estimates the impact of a mutation on the stability and function of the protein using structural and evolutionary analyses with amino acid substitution. The tool evaluates the mutation as probably damaging, possibly damaging, benign, or of unknown significance, using quantitative prediction with a score.

### 2.2. Bioinformatics Analysis of Gene Expression, Mirna Interactions, and Pharmacogenomics

Protein–protein interaction analysis of SARS-CoV-2-interacting proteins was carried out using the GeneMANIA and STRING web resources [15,16]. Single-cell RNA sequencing (RNA-seq) data were downloaded from the PanglaoDB database, which contains more than 1300 single-cell sequencing samples [17]. Lung single-cell RNA-seq data were obtained from the Sequence Read Archive [18] and processed in the R software environment using the Seurat package [19].

Analysis of the interactions of miRNAs with target proteins was performed using information from two databases: miRTarBase, which contains information from more than 8000 referenced sources about experimentally confirmed miRNA–protein interactions [20]; and miRPathDB, which contains experimentally confirmed and predicted miRNA–protein interactions [21]. Data on the differential expression of miRNAs in various cell cultures were downloaded from the database of the FANTOM5 project [22]. The DrugBank database [23] was searched for drugs that could change the expression levels of proteins.

## 3. Results and Discussion

### 3.1. Protein–Protein Interaction Networks of Sars-Cov-2-Interacting Genes

The protein–protein interaction networks obtained with two different tools GeneMANIA and STRING (Figure 1 and Figure 2) demonstrated that TMPRSS2 was co-expressed with other SARS-CoV-2-interacting genes, despite showing contradictory co-expression patterns. According to GeneMANIA, TMPRSS2 was co-expressed with BSG, whereas STRING indicated co-expression of ACE2 and TMPRSS2. Interestingly, BSG showed the maximum number of protein–protein interactions in both networks.

BSG, or extracellular matrix metalloproteinase inducer, also known as cluster of differentiation 147 (CD147), encoded by the BSG gene, is a transmembrane glycoprotein of the immunoglobulin superfamily and a determinant of the Ok blood group system. The BSG protein has an important role in targeting monocarboxylates (MCT) transporters SLC16A1, SLC16A3, SLC16A8, SLC16A11, and SLC16A12 to the plasma membrane by interaction with MCT molecules via its transmembrane and cytoplasmic domains.

BSG is involved in spermatogenesis, embryo implantation, neural network formation, and tumor progression. It stimulates adjacent fibroblasts to produce matrix metalloproteinases. BSG seems to be a receptor for oligomannosidic glycans and, according to in vitro experiments, can promote outgrowth of astrocytic processes [24,25,26]. BSG is also involved in tumor development, plasmodium invasion, and viral infection [27,28,29,30,31].

Previous data on SARS indicate that BSG has a functional role in facilitating SARS-CoV invasion of host cells, and CD147-antagonistic peptide-9 has a high binding rate to HEK293 cells and an inhibitory effect on SARS-CoV [32]. Based on the similarity of SARS-CoV and SARS-CoV-2, the function of BSG in invasion of host cells by SARS-CoV-2 can be assumed. The exact role of BSG in COVID-19 is still unknown; however, it was recently shown that CD147 may bind to the spike protein of SARS-CoV-2 [1]. Preliminary data from a small sample of COVID-19 patients demonstrated that meplazumab, a humanized anti-CD147 antibody, efficiently improved the recovery of patients with SARS-CoV-2 pneumonia and showed a favorable safety profile [33].

### 3.2. Expression of ACE2, BSG, and TMPRSS2 in Single Cells

Expression profiles of SARS-CoV-2-interacting genes in various tissues demonstrated that ACE2 had high level of expression only in testicles in peritubular myoid cells (Figure 3). The highest expression levels of BSG were found in germ cells, endothelia of various localizations, fibroblasts, and some other cell types (Figure 4). TMPRSS2 showed high levels of expression in the prostate, intestines, and lungs (Figure 5).

In addition, the expression levels of these proteins were analyzed in a single sample (SRS2769051) of proximal stromal lung cells (Figure 6, Figure 7, Figure 8 and Figure 9). ACE2 had low levels of expression in pulmonary alveolar cells and fibroblasts. BSG was characterized by average levels of expression in fibroblasts and alveolar cells. Only the TMPRSS2 gene demonstrated tissue-specific expression in alveolar cells. Given the high specificity of the expression of TMPRSS2 in lung tissue, we further studied genomic and epigenomic properties that may affect its expression levels and the affinity of its interactions with viral particles.

### 3.3. Snv and Indel Variants of the Tmprss2 Gene

According to information in the GnomAD database, 1025 SNVs and indels of various frequencies, functional impact, and localization have been described in the TMPRSS2 gene. This list includes 332 missense variants, 17 frameshifts, 64 splice site variants, 14 stop codon mutations, and three in-frame indels. Among frequent variants (minor allele frequency >0.01), there were only 13 intronic polymorphisms, five synonymous variants, and two missense mutations (rs12329760 and rs75603675). Both missense variants had high frequencies (24.8% and 35.0% in GnomAD, respectively) (Table 1).

The variant rs12329760 is a mutation of C to T in the 589 position of the gene, which leads to a change from valine to methionine at amino acid position 197 (exon 7) of transmembrane protease serine 2 isoform 1, or at position 160 (exon 6) of isoform 2. This mutation is predicted by Poly-Phen-2 to be probably damaging, with a score of 0.997 (sensitivity: 0.41; specificity: 0.98). The T allele of the TMPRSS2 rs12329760 variant was positively associated with TMPRSS2-ERG fusion by translocation; it was also associated with an increased risk of prostate cancer in European and Indian populations [34,35]. The rs75603675 variant (C to A transition in position 23, Gly8Val) was not reported to be associated with prostate cancer or any other clinical condition. An interesting feature of both frequent missense variants was the difference in prevalence between European and Asian populations; rs12329760 was 15% more frequent in populations of East Asia (38%) than in European populations (23%). For rs75603675, the difference was even more significant: the minor allele reached 42% in European populations and about 1% in populations of East Asia. Regarding CNVs, the controlDB database contained only one deletion in the TMPRSS2 gene (one copy variant) with relatively low frequency (1.2%) (Table 2).

The structural variability of the TMPRSS2 gene in relation to COVID-19 has recently been investigated by many research groups. In particular, Paniri et al. used various bioinformatics approaches to predict the functional consequences of TMPRSS2 SNPs with respect to susceptibility to SARS-CoV-2, the functional effects of SNPs on splicing, and the influence of polymorphisms on miRNA function. According to Paniri et al., rs12329760 showed the highest scores in various analyses and was considered deleterious by three tools, indicating its negative functional impact. On the contrary, rs75603675 (G8D) was considered deleterious only by polyphen-2, whereas other tools such as Phyre2, GOR IV, and PSIPRED predicted that both variants would have functional effects on the secondary structure of the TMPRSS2 protein [36].

### 3.4. Frequency of Protein-Changing Allelic Variants of the Tmprss2 Gene in Populations of North Eurasia

In order to study the population differentiation in TMPRSS2 functional variants in more detail, we searched for TMPRSS2 allele frequencies in our own unpublished data from 76 populations of North Eurasia, based on 1836 samples genotyped using genome-wide microarrays. Four missense mutations and two CNVs in the TMPRSS2 gene were found in our dataset. We compared the frequency of TMPRSS2 missense mutations in the North Eurasian population (Table 3) with the worldwide data (Table 4). Three missense mutations (rs148125094, rs143597099, and rs201093031) were very rare variants, whereas rs12329760, which was previously shown to be associated with prostate cancer, was found with high frequency in all populations. Data on the second high-frequency missense variant in the TMPRSS2 gene according to the GnomAD database (rs75603675) were not available because of the absence of this SNP from the microarray used in our study. The minor allele of variant rs148125094 was found on only two chromosomes (total frequency 0.00054) in single heterozygous individuals from the Karelian and Abkhaz populations. The variant rs143597099 was present only in one heterozygote from the Veps population. The variant rs201093031 was found in North-East Asian Nivkh and Udege populations with a frequency of 7%, and in a single Tuvan individual from Siberia. The frequency of the probably damaging minor allele of the rs12329760 polymorphism ranged from 10% (in the Khvarshi population from Dagestan) to 63% (in Sagays Khakas). In general, the minor allele T had higher frequencies in Siberia and Central Asia (both around 35%), whereas the lowest frequencies of the damaging variant were found in North Caucasus (19%), Dagestan (22%), and Eastern Europe (29%). This distribution is consistent with the worldwide data, which demonstrates a much higher frequency of the minor allele in Asian populations (36–41%) than in Europeans (22–24%) (Table 4).

In addition, we detected CNVs in two samples. In the first case, an increase in the number of copies covering the entire gene was found in a Karanogai individual. The second CNV, affecting exons 3–7, was found in a single Kumyk individual.

Thus, potentially functionally significant variants in the TMPRSS2 gene were found in many human populations with relatively high frequencies, demonstrating region-specific distribution patterns. Both variants—the probably damaging SNV and heterozygous deletion of the gene—may significantly affect the interactions of this human serine protease with viral spike proteins, thereby changing the efficacy of the priming of viral proteins by TMPRSS2. However, the roles of the TMPRSS2 gene and its variants in interactions with SARS-CoV-2 and in viral infectivity still need to be elucidated.

### 3.5. Regulation of Expression of Tmprss2

#### 3.5.1. Eqtls

According to the GTEx Analysis V8 database, the TMPRSS2 gene contains 136 eQTLs (including 60 downregulating and 76 upregulating variants) that significantly alter its expression in lung tissues (Table 5). However, in general, these eQTLs have only minor effects on gene expression. The average slope of the regression line (the value that characterizes the strength of an eQTL’s effect) was around 0.09 both for down- and upregulating variants. The strongest single variant could change the gene’s expression by 13%.

#### 3.5.2. Mirnas

According to the miRTarBase and miRPathDB databases, no experimentally proven miRNAs regulating TMPRSS2 have been detected. It is worth noting that the TMPRSS2 and BSG genes have the same predicted regulatory miRNAs.

The top 30 miRNAs predicted to regulate TMPRSS2 and BSG were analyzed for enrichment in various cell types using the FANTOM5 database. None of the top miRNAs was enriched in lung or respiratory tract cells, but three miRNAs showed slight expression in immune and endothelial cells (Table 6).

#### 3.5.3. Pharmacotranscriptomics of Tmprss2

According to the DrugBank database, nine drugs can reduce the level of expression of TMPRSS2. For five of them (acetaminophen/paracetamol, curcumin, cyclosporine, and ethinylestradiol), this effect has been clinically proved (Table 7). Information on the direction of the effect of estradiol is conflicting; in different experiments it has been found to either downregulate or upregulate TMPRSS2 expression.

Two drugs from the list above (acetaminophen/paracetamol and curcumin) have also been considered as possible therapies for COVID-19 [37]. Acetaminophen is currently being discussed as a possible drug for the correction of fever in patients with COVID-19. The ability of this drug to reduce the level of expression of TMPRSS2 may be an additional argument in favor of its use, compared with non-steroidal anti-inflammatory drugs. Curcumin, a widely used food supplement, has the predicted ability to block the main protease (Mpro) of SARS-CoV-2 [38] and may be studied further in relation to COVID-19 therapy. However, only pentanal, which enhances the expression of ACE2, is described in DrugBank as a drug that can change the expression level of ACE2. According to this database, expression of BSG1 is affected by eight drugs, five of which can reduce the level of the protein. One of them, valproic acid, can also reduce the expression of TMPRSS2 (Table 8).

## 4. Conclusions

The TMPRSS2 protein plays a crucial part in the process of SARS-CoV-2 activation in human cells. The gene encoding this protease demonstrates a high level of genetic variability, as well as having many variants that may regulate its expression levels. Although very few of the potentially functionally significant variants of this gene are of relatively high frequency, population-specific patterns of TMPRSS2 variability may contribute to some extent to the different viral infectivity of SARS-CoV-2 in populations of various geographic origins.

TMPRSS2 is probably co-expressed with SARS-CoV-2 receptors (ACE2 and BSG), but only the TMPRSS2 protease demonstrates tissue–specific expression in alveolar cells, the target cell type of SARS-CoV-2. Thus, TMPRSS2 is potentially the most promising target for COVID-19 therapy, based on its specific expression in lung, its important role in the process of cell infection, and its interactions with other proteins involved in the infection process. Several well-studied drugs can downregulate the expression of TMPRSS2 in human cells, including acetaminophen and curcumin. Both deserve close attention as possible anti-COVID-19 drugs, owing to their confirmed effects on TMPRSS2 expression, as well the long history of their use, their known side effects, and their wide availability.

## Figures and Tables

**Figure 1 genes-12-00019-f001:**
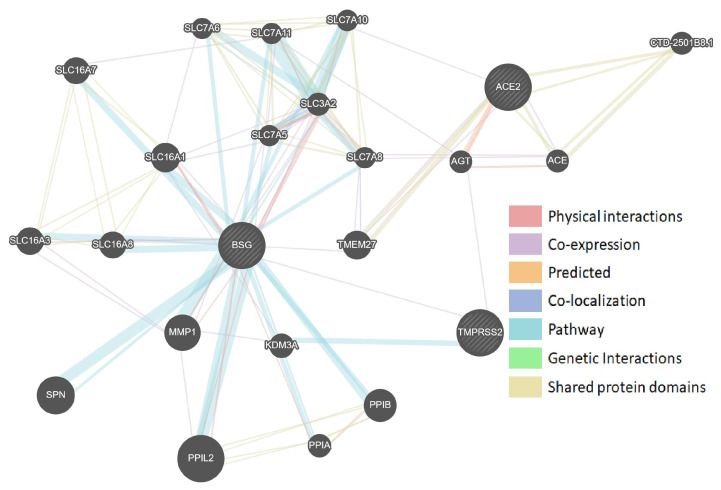
Protein–protein interaction network of SARS-CoV-2-interacting genes (GeneMANIA). The size of the circles indicates the strength of the bond with the gene being assessed.

**Figure 2 genes-12-00019-f002:**
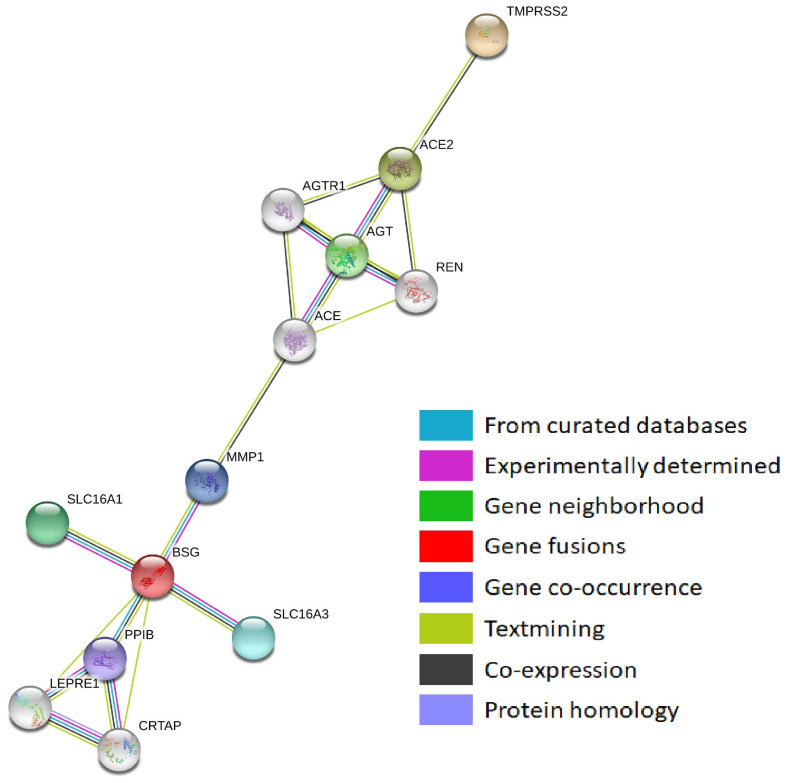
Protein–protein interaction network of SARS-CoV-2-interacting genes (STRING).

**Figure 3 genes-12-00019-f003:**
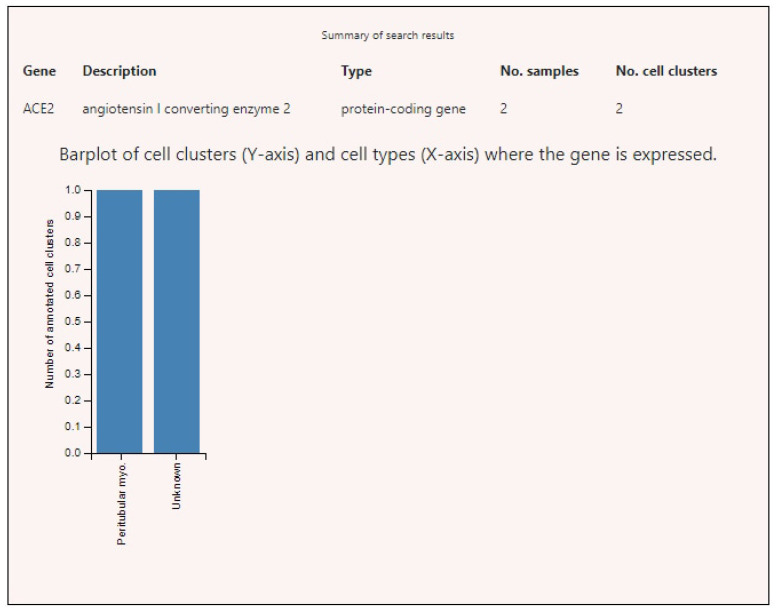
Angiotensin-converting enzyme 2 (ACE2) expression levels in various human cells. Note: Unknown cells of unknown types present in testicle tissue specimens.

**Figure 4 genes-12-00019-f004:**
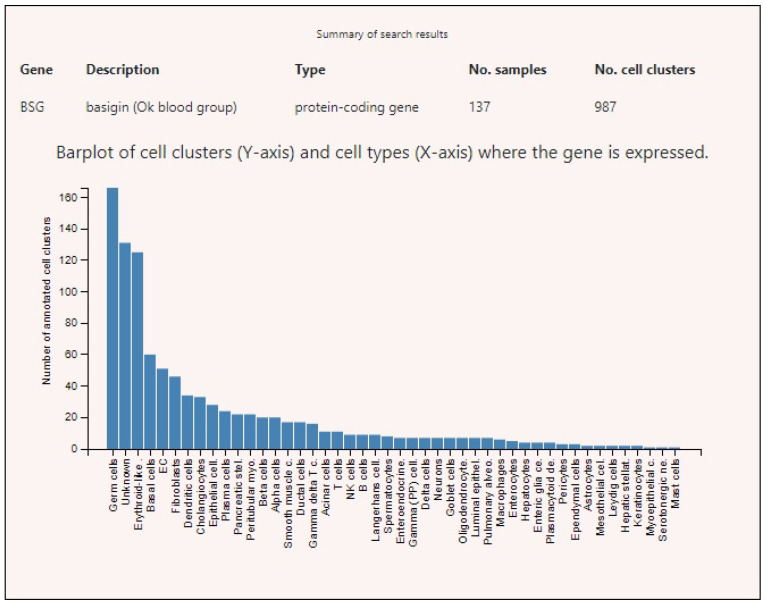
Basigin (BSG) expression levels in various human cells.

**Figure 5 genes-12-00019-f005:**
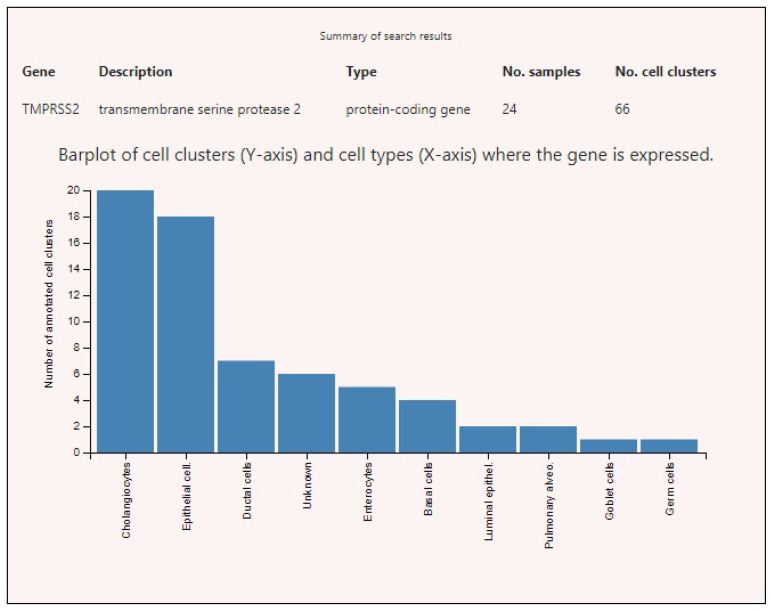
TMPRESS2 expression levels in various human cells.

**Figure 6 genes-12-00019-f006:**
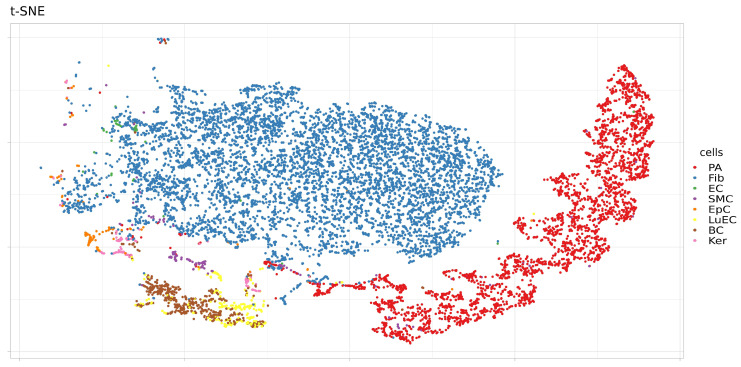
Cell types in a single sample (SRS2769051) of proximal stromal lung cells. PA, pulmonary alveolar type II cells; Fib, fibroblasts; EC, endothelial cells; SMC, smooth muscle cells; LuEC, luminal epithelial cells; BC, basal cells; Ker, keratinocytes. Color intensity indicates the level of expression in a single cell.

**Figure 7 genes-12-00019-f007:**
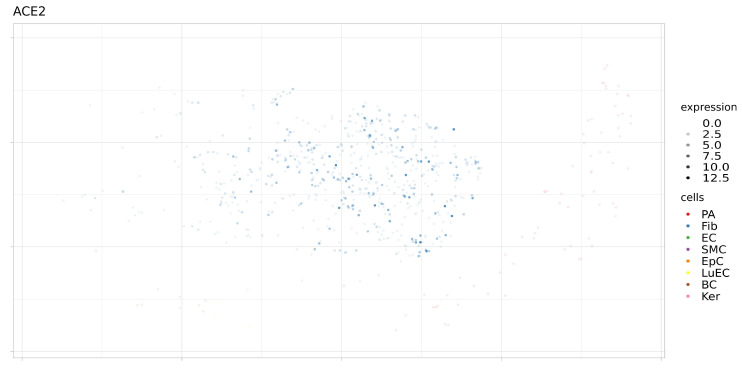
Expression levels ACE2 in separate cells of proximal stromal lung (SRS2769051). PA, pulmonary alveolar type II cells; Fib, fibroblasts; EC, endothelial cells; SMC, smooth muscle cells; LuEC, luminal epithelial cells; BC, basal cells; Ker, keratinocytes. Color intensity indicates the level of expression in a single cell.

**Figure 8 genes-12-00019-f008:**
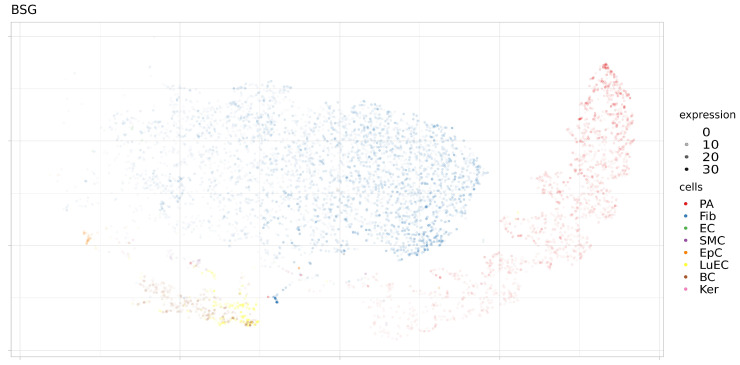
Expression levels BSG in separate cells of proximal stromal lung (SRS2769051). PA, pulmonary alveolar type II cells; Fib, fibroblasts; EC, endothelial cells; SMC, smooth muscle cells; LuEC, luminal epithelial cells; BC, basal cells; Ker, keratinocytes. Color intensity indicates the level of expression in a single cell.

**Figure 9 genes-12-00019-f009:**
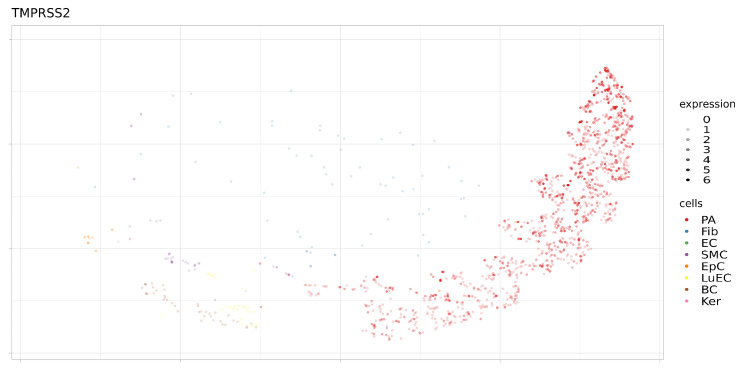
Expression levels TMPRSS2 in separate cells of proximal stromal lung (SRS2769051). PA, pulmonary alveolar type II cells; Fib, fibroblasts; EC, endothelial cells; SMC, smooth muscle cells; LuEC, luminal epithelial cells; BC, basal cells; Ker, keratinocytes. Color intensity indicates the level of expression in a single cell.

**Table 1 genes-12-00019-t001:** Single nucleotide variant (SNV) and INDEL variant in the TMPRSS2 gene.

Variant Type	Maf > 0.01	Maf < 0.01
3_prime_UTR_variant	0	22
5_prime_UTR_variant	0	5
frameshift_variant	0	17
inframe_deletion	0	1
inframe_insertion	0	2
intron_variant	13	409
missense_variant	2	332
splice_acceptor_variant	0	4
splice_donor_variant	0	5
splice_region_variant	0	54
stop_gained	0	13
stop_lost	0	1
synonymous_variant	5	140

**Table 2 genes-12-00019-t002:** Copy number variation (CNV) variants in the TMPRSS2 gene.

Region	N	Frequency %	Copy Number	Gene Name
Chr21:42857241-42863723	164	1.2195122	1	TMPRSS2

**Table 3 genes-12-00019-t003:** Frequency of TMPRSS2 missense mutations in North Eurasian population. AZ General information about the region from individual populations was highlighted in bold.

Population	N	rs148125094	rs143597099	rs12329760	rs201093031
**Easten Europe**	**419**	0.0012	0.0012	0.2983	0.0000
Bashkirs Burzyan	29	0.0000	0.0000	0.1552	0.0000
Bashkirs Perm	15	0.0000	0.0000	0.3000	0.0000
Bashkirs Salavat	15	0.0000	0.0000	0.3667	0.0000
Besermians	16	0.0000	0.0000	0.1563	0.0000
Chuvash	26	0.0000	0.0000	0.3077	0.0000
Karelians	29	0.0172	0.0000	0.3966	0.0000
Komi	30	0.0000	0.0000	0.3333	0.0000
Mari	30	0.0000	0.0000	0.2500	0.0000
Mordvins Erzya	16	0.0000	0.0000	0.3125	0.0000
Mordvins Moksha	30	0.0000	0.0000	0.3000	0.0000
Mordvins Shoksha	14	0.0000	0.0000	0.3214	0.0000
Russians	33	0.0000	0.0000	0.3333	0.0000
Tatars Kazan	30	0.0000	0.0000	0.3000	0.0000
Udmurts	30	0.0000	0.0000	0.3000	0.0000
Udmurts Balezino	28	0.0000	0.0000	0.3214	0.0000
Udmurts Sharkan	18	0.0000	0.0000	0.2500	0.0000
Veps	30	0.0000	0.0167	0.3333	0.0000
**North Caucasus (excl. Dagestan)**	**274**	0.0018	0.0000	0.1989	0.0000
Abkhaz	30	0.0167	0.0000	0.3000	0.0000
Adyghe	10	0.0000	0.0000	0.1500	0.0000
Balkars	50	0.0000	0.0000	0.1800	0.0000
Chechens	27	0.0000	0.0000	0.2222	0.0000
Cherkess	30	0.0000	0.0000	0.2167	0.0000
Ingush	30	0.0000	0.0000	0.1500	0.0000
Karachays	22	0.0000	0.0000	0.2045	0.0000
Mingrelians	28	0.0000	0.0000	0.1607	0.0000
North Ossetians	30	0.0000	0.0000	0.1833	0.0000
South Ossetians	17	0.0000	0.0000	0.2059	0.0000
**Dagestan**	**538**	0.0000	0.0000	0.2309	0.0000
Aghuls	24	0.0000	0.0000	0.2292	0.0000
Akhvakhs	24	0.0000	0.0000	0.3125	0.0000
Andis	17	0.0000	0.0000	0.2353	0.0000
Archins	24	0.0000	0.0000	0.3333	0.0000
Avars	24	0.0000	0.0000	0.1875	0.0000
Bagulals	23	0.0000	0.0000	0.3261	0.0000
Bezhtins	22	0.0000	0.0000	0.2273	0.0000
Botlikhs	16	0.0000	0.0000	0.1250	0.0000
Chamalals	24	0.0000	0.0000	0.2083	0.0000
Dargins	28	0.0000	0.0000	0.2321	0.0000
Ginukhs	19	0.0000	0.0000	0.0000	0.0000
Gunzibians	17	0.0000	0.0000	0.0294	0.0000
Karanogais	19	0.0000	0.0000	0.2368	0.0000
Karatins	24	0.0000	0.0000	0.3333	0.0000
Khvarshins	15	0.0000	0.0000	0.1000	0.0000
Kumyks	37	0.0000	0.0000	0.2703	0.0000
Laks	24	0.0000	0.0000	0.3125	0.0000
Lezgins	28	0.0000	0.0000	0.2037	0.0000
Nogais	20	0.0000	0.0000	0.2750	0.0000
Rutuls	22	0.0000	0.0000	0.1818	0.0000
Tabasarans	21	0.0000	0.0000	0.2619	0.0000
Tindins	18	0.0000	0.0000	0.2222	0.0000
Tsakhurs	24	0.0000	0.0000	0.2292	0.0000
Tsez	24	0.0000	0.0000	0.2708	0.0000
**Central Asia**	**128**	0.0000	0.0000	0.3565	0.0000
Dungans	23	0.0000	0.0000	0.4130	0.0000
Kazakh Junior Horde	29	0.0000	0.0000	0.2931	0.0000
Kazakh Great Horde	26	0.0000	0.0000	0.4423	0.0000
Kyrgyz	28	0.0000	0.0000	0.3704	0.0000
Uzbeks	22	0.0000	0.0000	0.2619	0.0000
**Siberia**	**404**	0.0000	0.0000	0.3540	0.0013
Altaians Maymalar	24	0.0000	0.0000	0.3958	0.0000
Altaians Kizhi	25	0.0000	0.0000	0.3600	0.0000
Buryats Aginskoe	23	0.0000	0.0000	0.4130	0.0000
Buryats Kurumkan	28	0.0000	0.0000	0.3929	0.0000
Chulyms	22	0.0000	0.0000	0.3636	0.0000
Evenks Yakutia	28	0.0000	0.0000	0.2857	0.0000
Evenks Zabaykalsky Krai	25	0.0000	0.0000	0.3200	0.0000
Kalmyks	29	0.0000	0.0000	0.3103	0.0000
Kets	15	0.0000	0.0000	0.3333	0.0000
Khakas Kachins	26	0.0000	0.0000	0.4423	0.0000
Khakas Sagays	29	0.0000	0.0000	0.6379	0.0000
Khanty Kazym	30	0.0000	0.0000	0.1333	0.0000
Khanty Russkinskie	26	0.0000	0.0000	0.2500	0.0000
Tomsk Tatas	20	0.0000	0.0000	0.3250	0.0000
Tuvans	28	0.0000	0.0000	0.3036	0.0185
Yakuts	26	0.0000	0.0000	0.4038	0.0000
**North East Asia**	**73**	0.0000	0.0000	0.2671	0.0284
Chukchi	25	0.0000	0.0000	0.3000	0.0000
Koryaks	20	0.0000	0.0000	0.3500	0.0000
Nivkhs	13	0.0000	0.0000	0.1538	0.0769
Udege	15	0.0000	0.0000	0.2000	0.0714

**Table 4 genes-12-00019-t004:** Frequency of TMPRSS2 missense mutations in worldwide data (GnomAD). AZ General information about the region from individual populations was highlighted in bold.

	rs148125094		rs12329760		rs201093031	
**Population**	**N**	**Frequency**	**N**	**Frequency**	**N**	**Frequency**
**European**	77147	0.0014	76846	0.2549	77117	0.0000
Finnish	12560	0.0016	12544	0.3725	12561	0.0000
Estonian	2416	0.0060	2394	0.3074	2406	0.0000
Southern European	5805	0.0013	5778	0.1748	5802	0.0000
North-western European	25402	0.0012	25348	0.2212	25391	0.0000
Other non-Finnish European	16562	0.0012	16439	0.2286	16557	0.0000
Swedish	13067	0.0010	13013	0.2722	13066	0.0000
Bulgarian	1335	0.0007	1330	0.1970	1334	0.0000
**South Asian**	15308	0.0007	15298	0.2477	15303	0.0000
**Latino**	17718	0.0003	17705	0.1533	17697	0.0000
**African**	12480	0.0002	12448	0.2918	12480	0.0000
**Ashkenazi Jewish**	5185	0.0000	5163	0.1352	5179	0.0000
**East Asian**	9196	0.0000	9188	0.3810	9193	0.0024
Japanese	76	0.0000	76	0.4013	76	0.0000
Korean	1909	0.0000	1909	0.3675	1909	0.0018
Other East Asian	7211	0.0000	7203	0.3844	7208	0.0026

**Table 5 genes-12-00019-t005:** Expression quantitative trait locus (eQTL) altering expression level TMPRSS2.

	N SNP	Average Maf	Average Slope
down	60	0.3722896	−0.09795966
up	76	0.4537386	0.09709619

**Table 6 genes-12-00019-t006:** MicroRNA regulating the expression of TMPRSS2 and BSG genes and showed expression in immune and endothelial cells. According to FANTOM5.

miRNA	Cell Ontology
hsa-miR-4476	B cell
hsa-miR-5187-3p	myeloid leukocyte
hsa-miR-5187-3p	hematopoietic cell
hsa-miR-7849-3p	endothelial cell
hsa-miR-7849-3p	blood vessel endothelial cell
hsa-miR-7849-3p	endothelial cell of vascular tree
hsa-miR-7849-3p	neutrophil

**Table 7 genes-12-00019-t007:** Drugs that alter expression TMPRSS2.

Drug	Drug Groups	Change	References
Acetaminophen	Approved	downregulated	21420995
Acyline	Investigational	downregulated	17510436
Stanolone	Illicit Investigational	downregulated	12711008
Stanolone	Illicit Investigational	upregulated	20601956, 23708653
Estradiol	Approved Investigational Vet Approved	downregulated	24758408
Estradiol	Approved Investigational Vet Approved	upregulated	19619570
Curcumin	Approved Experimental Investigational	downregulated	18719366, 22258452
Cyclosporine	Approved Investigational Vet Approved	downregulated	20106945
Calcitriol	Approved Nutraceutical	upregulated	21592394, 26485663
Entinostat	Investigational	upregulated	26272509
Ethinylestradiol	Approved	downregulated	18936297
Genistein	Investigational	downregulated	15378649, 26865667
Metribolone	Experimental	downregulated	12711008
Metribolone	Experimental	upregulated	17010675, 21440447
Resveratrol	Investigational	downregulated	18586690
Selenium	Approved Investigational Vet Approved	upregulated	19244175
Testosterone	Approved Investigational	upregulated	21592394
Tretinoin	Approved Investigational Nutraceutical	upregulated	23830798
Valproic acid	Approved Investigational	upregulated	23179753, 24383497, 26272509
Zoledronic acid	Approved	upregulated	24714768

**Table 8 genes-12-00019-t008:** Drugs that alter expression BSG.

Drug	Drug Groups	Change	References
Amiodarone	Approved Investigational	upregulated	19774075
Arsenic trioxide	Approved Investigational	downregulated	23232515
Estradiol	Approved Investigational Vet Approved	upregulated	19167446
Methotrexate	Approved	downregulated	25339124
Quercetin	Experimental Investigational	upregulated	21632981
Isotretinoin	Approved	downregulated	20436886
Silicon dioxide	Approved	downregulated	25895662
Valproic acid	Approved Investigational	downregulated	23179753

## Data Availability

Publicly available datasets were analyzed in this study. This data can be found here: https://panglaodb.se/, https://gnomad.broadinstitute.org/, https://fantom.gsc.riken.jp/5/, https://gtexportal.org/home/datasets.

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
