# Peer review of "Structural Variability, Expression Profile, and Pharmacogenetic Properties of TMPRSS2 Gene as a Potential Target for COVID-19 Therapy"

_genes, 2020, doi:10.3390/genes12010019_

Round 1
Reviewer 1 Report
General Comments
The manuscript describes the potential of TMPRSS2 as a potential drug target for COVID-19 based on structural information, cell and tissue-specific expression and identification of gene variations of TMPRSS2 in 76 populations in North Eurasia. Generally, the manuscript is clearly designed and relatively well written. However, before being accepted for publication, the authors are requested to revise it taking into account the comments/suggestions presented below.
Why is the Results and Discussion section presented before Materials and Methods? This is not according to the instructions for the journal.
The quality of Figs 7-9 is poor. Although I understand that the expression levels are low, it is really hard to get any indication based on the figures.
Specific Comments
Abstract
P1, L2: Delete “novel”, it is already a year since the outbreak started in China
Introduction
P1, L1: “Novel” > “The”
P1, L4: Define “ACE”
P2, L1: Define “BSG”
P2, L2: “P” in TMPRSS2” already stands for protease
P2, L9: “microRNAs; miRNAs” > “microRNAs (miRNAs)”
P2, L12 & 13: Define “ETS”, “ERG” and “ETV1”
P2, L19: “coronavirus of the” > “coronavirus responsible for”
P2, L20: “In this work” > “Here”
Results and Discussion
P2, L4: “co-expression between” > “co-expression of”
Figures 1 and 2
P2: The text in both figures is too small
P3, L4: “targeting......to the plasma membrane” is not clear
2.2
P3, L2: “a high level” > “high level”
Fig. 3: What is the message in the figure? In the text it is mentioned that high level expression is seen only in testicles, but “unknown” shows the same level of cell clusters. This does not make sense.
Figs 4 & 5: The text in the figures is too small
Author Response
Thank you for your comments and suggestions.
We accepted all your suggestions. The revised text can be found in the attached file.

Reviewer 2 Report
This piece is relavant and interesting for readers.
There are, however, some corrections to be made, listed below.
Abstract
Line 1 The human serine protease serine 2 (TMPRSS2) instead of
The human serine protease serine 2 TMPRSS2
Line 4 Angiotensin-converting enzyme 2 (ACE-2) instead of
ACE-2
Line 4 Basigin ( BSG ) instead of BSG
Keywords: Also put ACE-2 and BSG.
Introduction
Pag 2 Line 8 : Delete one of the two brackets : (SNVs),indels)
Results and Discussion
Pag 2 Line 27 and 28 : , GeneMANIA and STRING (Figs.1,2), instead of
( GeneMANIA and STRING ) (Figs. 1,2)
Pag 3 Line 1 : BSG, or extracellular matrix metalloproteinase inducer (EMMPRIN) instead of Basigin, or extracellular matrix metalloproteinase inducer
References
For references, follow in Genes
Instructions for Authors:
Quick Reference Formatting Guide, indicating the DOI in all articles, if any, as indicated in red below:
- Bowman, C.M.; Landee, F.A.; Reslock, M.A. Chemically Oriented Storage and Retrieval System. 1. Storage and Verification of Structural Information. J. Chem. Doc.1967, 7, 43-47; DOI:10.1021/c160024a013
Reference 24.
Paniri, A.; Hosseini, M.M.; Akhavan-Niaki, H. First comprehensive computational analysis of functional consequences of TMPRSS2 SNPs in susceptibility to SARS-CoV-2 among different populations. Journal of Biomolecular Structure and Dynamics 2020, 1–18; DOI: 10.1080/07391102.2020.1767690.
Instead of
Paniri, A.; Hosseini, M.M.; Akhavan-Niaki, H. First comprehensive computational analysis of functional consequences of TMPRSS2 SNPs in susceptibility to SARS-CoV-2 among different populations. Journal of Biomolecular Structure and Dynamics 2020, pp. 1–18.
Reference 31 should be replaced with a more recent analogue
Author Response

(The authors gave the same response as above.)
